# A Real-Time Control Method for Upper Limb Exoskeleton Based on Active Torque Prediction Model

**DOI:** 10.3390/bioengineering10121441

**Published:** 2023-12-18

**Authors:** Sujiao Li, Lei Zhang, Qiaoling Meng, Hongliu Yu

**Affiliations:** 1Institute of Rehabilitation Engineering and Technology, University of Shanghai for Science and Technology, Shanghai 200093, China; sujiaoli2015@foxmail.com (S.L.); 15162875837@163.com (L.Z.); qiaoling_meng@126.com (Q.M.); 2Shanghai Engineering Research Center of Assistive Devices, Shanghai 200093, China; 3Key Laboratory of Neural-Functional Information and Rehabilitation Engineering of the Ministry of Civil Affairs, Shanghai 200093, China

**Keywords:** sEMG, real-time control, active torque prediction model, rehabilitation exoskeleton

## Abstract

Exoskeleton rehabilitation robots have been widely used in the rehabilitation treatment of stroke patients. Clinical studies confirmed that rehabilitation training with active movement intentions could improve the effectiveness of rehabilitation treatment significantly. This research proposes a real-time control method for an upper limb exoskeleton based on the active torque prediction model. To fulfill the goal of individualized and precise rehabilitation, this method has an adjustable parameter assist ratio that can change the strength of the assist torque under the same conditions. In this study, upper limb muscles’ EMG signals and elbow angle were chosen as the sources of control signals. The active torque prediction model was then trained using a BP neural network after appropriately extracting features. The model exhibited good accuracy on PC and embedded systems, according to the experimental results. In the embedded system, the *RMSE* of this model was 0.1956 N·m and 94.98%. In addition, the proposed real-time control system also had an extremely low delay of only 40 ms, which would significantly increase the adaptability of human–computer interactions.

## 1. Introduction

Stroke is a typical disease of brain function disorder, caused by acute cerebrovascular injury [1]. Hemiplegia, facial paralysis, vision problems, and linguistic disturbances are generally considered as common sequelae. Among these, hemiplegia is the most typical symptom [2]. The upper limb is one of the most frequently used systems in activities of daily life (ADL), which has a significant impact on quality of life (QOL) [3]. According to reports, the disabilities brought on by a stroke often result in significantly lower QOL compared with that of their contemporaries. In order to enhance QOL, it is crucial for stroke patients, their families, and society to assist them in regaining lost motor function [4].

According to clinical experiences, the recovery of dysfunction is crucial at all stages of neurological impairment. Continuous movement and relearning strengthen and consolidate the nervous system’s plasticity [2]. In the treatment of post-stroke patients, motor relearning is one of the primarily used rehabilitation strategies [5]. Based on the neuromuscular system’s plasticity and capability for learning, repeated correct exercise training can encourage it to regain lost abilities [3].

Traditional rehabilitation therapy, on the other hand, entails repetitive one-on-one patient training and has a number of drawbacks, including high rehabilitation training expenditures and low rehabilitation effectiveness [6]. At the same time, a large number of patients are unable to receive appropriate rehabilitation and the opportunity for rehabilitation due to limited medical resources and shortages of rehabilitation physiotherapists [7]. Robotic rehabilitation techniques are being developed to address the issues and pain points presenting in the current rehabilitation therapy process [8], which has the potential to significantly increase the effectiveness and efficiency of rehabilitation training [9]. When performing physical treatments after a stroke, a variety of rehabilitation robots can primarily offer the benefits of being intensive, accurate, quantitative, and safe when compared with traditional “one-to-one” manual physical therapies [10]. However, there are some deficiencies of traditional rehabilitation robots, such as complex structures, poor compliance, and poor portability, hindering the application of more rehabilitation scenarios and stages [11]. Wearable exoskeletons, designed with electro-mechanical systems, could be employed to assist, augment, or enhance motion and mobility in a variety of human motion applications and scenarios, even in unconventional rehabilitation settings, for user-independent self-help rehabilitation (e.g., home) [12].

Passive rehabilitation training relying on exoskeleton rehabilitation robots, limited by the active engagement of patients, presented an unsatisfactory effect of dysfunction recovery. [13]. Exoskeleton rehabilitation robots require better interaction performance to promote the effect of rehabilitation training [14]. Electromyography (EMG) signals captured from muscle surfaces, as neuromuscular signals produced by the muscles that drive movement, are used as input signals in some exoskeleton designs [15]. To create a strong and intelligent control system, as closely as feasible to the human neuromuscular control system, many studies have tried to connect EMG signals to forces or torques produced by muscles [16,17,18].

Several studies have exploited EMG signals from contraction muscles to assess joint torque and applied auxiliary joint torque to offer additional strength dependent on the magnitude of the active torque. The studies demonstrated that an exoskeleton rehabilitation robot with active intention control improved the elbow torque function of stroke patients. Conversely, the robotic system would not provide auxiliary torque if EMG signals motivating the movements were not activated [4]. Therefore, all subjects were urged to actively engage in exercise training by using their EMG signals [4]. More engagement during the movement can be provided by active intention control via continuous EMG signals, which is advantageous to assist stroke patients in regaining motor function [4].

Previous studies have proposed many rehabilitation robots driven by user intentions since the first robotic system for upper limb rehabilitation training, known as MIT-MANUS, was created by MIT researchers in 1991 [12]. Some essential drive control models used in the context of the exoskeleton mechanism have been briefly reviewed and discussed. Zhang developed a 3-DOF upper limb exoskeleton that was controlled to let the patient move based on their intention. He collected the properties of EMG signals from the time, frequency, and time–frequency domains in order to precisely extract the patients’ motor intentions from the signals. He then trained a continuous motion control model using the back propagation (BP) neural network. The outcomes demonstrated this model presented a minor angle estimation error [19]. To replicate elbow motion, Wang et al. established a robotic joint with one degree of freedom and gathered EMG signals as biofeedback signals. The intention of flexion and extension motions was quantitatively presented by evaluating the collected EMG data, and then online control instructions for the exoskeleton were created. Given this, the exoskeleton can actively assist in elbow joint movement in accordance with the intention of flexion and extension movements [3]. Song et al. created a robotic arm with 1-DOF that is controlled by EMG signals in order to help stroke patients to exercise their elbows in the horizontal plane, which can provide continuous extension torque assistance and reverse torque during movement that is proportional to the amplitude of the subject’s triceps muscle’s EMG signals [4]. By using EMG signals from the biceps and triceps muscles to operate the exoskeleton for elbow abduction and adduction, Yahya Z et al. presented a design for a powered elbow exoskeleton for helping elbow movement [16]. A low-cost 1-DOF exoskeleton was created by Anand Asokan et al. using EMG signals from the biceps and triceps brachii muscles as the control signal sources. They converted the EMG data from each muscle into binary signals, and then designed the dual binary input or gate motor control logic, a straightforward control method that required a lower amount of hardware [20]. Qingcong Wu et al. constructed an adaptive neural cooperative controller for an upper limb rehabilitation robot, which employed interaction forces and EMG data from the biceps and triceps muscles as inputs to enable patients to perform intention-based cooperative rehabilitation training. In this approach, biological forces and interaction forces were mapped to incremental displacements that represented human intention [21].

To enhance their motor function as much as possible, patients should be able to perform exercise training not only at medical institutions, but also at home. Some studies have suggested that home-based robotic therapy is a trend in post-stroke upper limb rehabilitation [22]. Therefore, we developed a single active degree of freedom exoskeleton, only 1.5 kg in weight, that can assist patients with elbow flexion and extension movements and would be easy enough to use at home, as well as in medical facilities and community centers. Based on the active freedom exoskeleton, a real-time active torque-prediction control model was proposed, which aims to provide auxiliary torque based on the intention of patients and assist patients in completing rehabilitation training, so as to improve the treatment effect significantly. We employed the BP neural network with momentum mechanism and adaptive learning algorithm for training active torque prediction models out of all the available techniques. To increase the model’s robustness, we incorporated multi-source physical and physiological signals as model inputs, such as elbow angle signals and EMG signals from several different muscles.

## 2. Materials and Methods

### 2.1. Upper Limb Rehabilitation Exoskeleton

For flexibility, compliance, and efficacy performance, an exoskeleton upper robot must be designed and validated to have high accuracy in control, be light in weight, and be reasonable in motion space in this study [23]. Thus, an adjustable design is necessary for upper limb rehabilitation robots when creating flexible, wearable solutions for the majority of patients [2]. In this study, the range of flexion and extension of the exoskeleton elbow joint was designed to be 0 to 120 degrees according to the degrees of freedom of the human upper limb elbow joint. According to the Human Dimensions of Chinese Adults (GB10000-1988) [24], the length of the upper arm and forearm of the exoskeleton is adjusted to meet the needs of different users. The three-dimensional model of the exoskeleton and Exoskeleton prototype are displayed in Figure 1 and Figure 2. In order to reduce the overall weight and ensure the strength of the instrument, the exoskeleton prototype was developed with a lightweight design, mostly made of carbon fiber and ABS, and weighed around 1.5 kg overall. Meanwhile, the exoskeleton adopted the design idea of the motor side and synchronous belts to transfer the torque to the elbow joint [25]. The side position of the motor reduced the volume, and the elbow joint and synchronous belt transmission provided steady and effective duplex torque transmission. Moreover, a synchronous belt tensioning system was also created for the exoskeleton to increase its service life.

### 2.2. Subject Information

Seven able-bodied male participants were recruited for this study. To prevent muscular tiredness from causing inaccurate experimental data and maintain the validity of the experiment, each subject was required to refrain from hard exercise 24 h before attending the experiment. Prior to commencing the experiment, anthropometric data of participants were collected and counted. The specific information of all subjects is shown in Table 1, including age (26 ± 2 years), height (173 ± 7 cm), weight (66.5 ± 13 kg), upper arm length (30.9 ± 4.9 cm), forearm length (26.45 ± 1.45 cm), hand length (18.7 ± 0.7 cm), and mean values of all subject parameters.

### 2.3. Data Acquisition

EMG and attitude sensors were used to collect the elbow joint angles and the EMG signals from multiple muscles of the upper limb, which are responsible for elbow flexion and extension action and driving the real-time control of the upper limb exoskeleton. The EMG sensor was a surface electromyographic signal sensor developed in collaboration with DFRobot and OYMotion. The attitude sensor uses the JY61 attitude sensor developed by Shenzhen Weite Intelligent Technology Co., Shenzhen, China. Based on the contribution analysis of muscles to elbow flexion and extension movements, the biceps brachii, triceps brachii, and brachioradialis muscle groups of the upper limb, which are responsible for elbow flexion and extension action, were chosen to drive the real-time control of the upper limb exoskeleton. Two attitude sensors were placed on the upper arm and forearm, respectively, to measure the angle at the elbow joint. The placement of the sensor is depicted in Figure 3. All the EMG signal and elbow angle acquisition sensors link to a signal-collecting control panel, which was responsible for controlling the frequency of the collected signals and sending them to the main control board via Bluetooth. The signal-collecting control panel adopted the STM32F103C8T6 chip. The main control board chip was the STM32F407ZGT6 chip developed by STMicroelectronics. The signal-collecting control panel receives data from EMG and attitude sensors. Then, the signal-collecting control panel transmits the data to the main control board through the Bluetooth module. The main control board processes the received signals and imports them into the elbow joint active torque prediction model. Figure 4 illustrates the data acquisition system.

According to DeLuca’s research, the amplitude of EMG signals is typically between 0.1 and 5 mV, the effective frequency ranges from 10 to 500 Hz, and the primary frequency falls between 50 and 150 Hz [26]. In the sampling theorem, EMG signals should be sampled at least twice as frequently as their effective frequency. In order to ensure the validity of the collected signals, the sample frequency used in this study was 1200 Hz for EMG signals and 100 Hz for elbow angle [27].

### 2.4. Data Processing and Feature Extraction

As the EMG signals were extremely weak and vulnerable to external interference, how to extract useful information from the acquired signals was the key to EMG signal processing. Before the feature extraction of the EMG signal, a 20–500 Hz bandpass Butterworth filter and a 50 Hz notch filter were used to remove undesirable outside noise and 50 Hz alternating current (AC) power interference, respectively. Generally, in order to improve the control effort of EMG signals, the features of the raw EMG signal must be extracted before serving as the input signal to the controller [28]. In this study, the root-mean-square (*RMS*) and integrated EMG (*iEMG*) of EMG signals were selected as characteristic values. Their calculation formulas were as follows,
(1)$$RMS=\sqrt{\frac{1}{N}\sum_{i=1}^{N}x_{i}^{2}}$$
(2)$$iEMG=\frac{1}{N}\sum_{i=1}^{N}\left|x_{i}\right|$$
where *N* is the number of the elbow joint torque samples and *x_i_* is the voltage at *i*th sampling. The EMG signals were non-stationary, which reduced their accuracy and dependability when used as motion-control signals. Thus, this work extracted angular velocity information from the elbow joint angle and used the pre-processed elbow joint angle and angular velocity as input characteristics in order to improve the robustness of the control system.

### 2.5. Multi-Modality Information Fusion

In this study, elbow angle and angular velocity were combined with the *iEMG* and *RMS* of three-channel EMG signals to create an eight-dimensional feature vector. F1–F5 represent the *RMS* of the biceps, triceps, and brachioradialis muscles, elbow joint angle, and angular velocity. F6–F8 represent the *iEMG* of the biceps, triceps, and brachioradialis muscles. In order to integrate the feature values of the EMG and angle signals, eliminate redundant information, and minimize matrix size, the extracted feature vectors were normalized and principal component analyses were performed for feature fusion. The contribution rate and cumulative contribution rate of each feature vector were computed, and the results are displayed in Table 2.

As can be seen from Table 1, the principal components above Eigenvector F5 following feature fusion by principal component analysis did not significantly improve the cumulative contribution rate of the feature vectors, whereas the first five-dimensional vectors represented 99.98% of all eight-dimensional feature vectors. Thus, the present optimized five-dimensional eigenvector reduced the dimensions of the feature matrix effectively.

### 2.6. Active Torque Prediction Model

This study compares the Hill model with the muscle torque model of the human upper limb bending process to quantitatively analyze the changes in elbow joint muscle force, providing a theoretical basis for upper limb muscle force training and making the training more targeted and efficient.

The Hill model is one of the classic muscle models and is widely used in biomechanical research. Based on the Hill model, this study establishes a mathematical model of the muscles [29]. As shown in Figure 5, this model considers muscles as a muscle model composed of a series elastic element (SEE), a passive elastic element (PEE), a contraction element (CE), and a feather-shaped horn (*φ*). Among them, *l_mt_* and *l_m_* respectively represent muscle length and muscle fiber length, while *l*_*t*1_ and *l*_*t*2_ represent tendon length.

After processing the original electromyographic signal and obtaining muscle activation, muscle force can be calculated using the following formula:$$l_{mt}=l_{m}\cos\phi +l_{t}=l_{m}\cos\phi +l_{t1}+l_{t2}$$
(3)$$F_{M}=(F_{CE}+F_{PE})\cos\phi$$
$$F_{CE}=af_{1}f_{v}F_{0}$$
$$a(t)=\frac{e^{Au(t)}-1}{e^{A}-1}$$
where *F_CE_* and *F_PE_* are the active and passive forces of the muscle fibers, respectively. Owing to the fact that the adaptive assistance control of the elbow joint in this study relies on the active torque of the elbow joint, only calculating *F_CE_* can meet the requirements of this study. The influencing factors of muscle fiber length and velocity *f*_1_, *f*_*v*_ were all calculated using the Thelen model in this study. *u*(*t*) is the preprocessed EMG signal, and *A* is the nonlinear influence factor.

The muscle torque model of the human upper limb bending process uses the second type of Lagrange method to calculate the elbow joint torque during the bending and extension of the human upper limb [30].
(4)$$T=m{\left(l_{3}+r_{2}\right)}^{2}\beta^{\prime \prime }-mg(l_{3}+r_{2})\sin\beta$$
where *m* is the mass and *g* is the acceleration of gravity, and in this study, 9.8 N/kg is considered. The length of the upper arm is the distance *l*_3_ from the center of mass of the upper arm to the center of rotation, which is the length of the forearm plus the radius *r*_2_ representing the elbow joint cylinder. The bending angle of the elbow joint is *β*.

From Figure 6, it can be seen that the trend of torque changes in the elbow joint calculated by the two methods was basically similar, both showing an increase followed by a decrease. In this article, the torque value calculated by the muscle torque model was higher than that of the Hill model. This was because the model ignored the output of other muscles in the elbow joint, and the biceps, triceps, and radial muscles shared the total muscle force, causing each muscle to bear a greater force. The small results calculated by the Hill model in this study were due to the fact that the adaptive assistance control of the elbow joint in this article relied on the active torque of the elbow joint, so only the active torque in the Hill model was calculated. Overall, the muscle torque model established in this study is still effective and feasible in calculating muscle force, so its results were taken as the expected output of the BP neural network.

Artificial neural networks (ANNs) have attracted increasing attention in the field of predictive modeling since they can create nonlinear relationships in high-dimensional data sets [31]. Moreover, the BP neural network model has overwhelming superiority in terms of robustness, memory ability, and the mapping ability of nonlinear relationships [32]. Therefore, we adopted the BP neural network to establish a prediction model for active torque at the elbow joint in this study. The layers of a BP neural network were classified into three types: input, hidden, and output layers. There was no link between the neurons in each layer, but there was a complete connection between the layers. In general, increasing the number of hidden layers can aid in improving the accuracy and reducing the network error, whereas it also complicates the network and can even cause overfitting. This paper uses a three-layer BP neural network to construct an active torque prediction model as it can actualize any mapping from *n* dimensions to *m* dimensions [33].

In the three-layer BP neural network, the five neural nodes of the input layer corresponded to the five-dimensional feature vectors that underwent principal component analysis. The node in the output layer represented the elbow torque, and there was one hidden layer with three hidden nodes; the complete network structure is displayed in Figure 7. As joint moments of human motion cannot be directly measured, many researchers compute joint moments based on the inverse dynamics of the human body [34]. Using the Lagrange equation to establish the dynamic equations of the mass point system in the form of generalized coordinates substantially simplified the solution procedure [35]. Thus, the second Lagrange equation was used to calculate the elbow torque during the bending and extension of the human upper limbs as the expected output of the BP neural network. In this proposed model, we employed the tansig function as the activation function of the hidden and output layers, the principal components *F*_1_–*F*_5_ as the input features, and *τ* as the predicted active elbow torque. The maximum number of training sessions was 3000, the training accuracy was 10^−3^, and the initial learning rate was set at 0.01. The output formula of this model is
(5)$$\tau =\tan sig[\omega_{out}\tan sig(\omega_{in}s+b_{in})+b_{out}]$$
(6)$$\tan sig=\frac{2}{1+e^{-2x}}-1$$
where *s* = [*F*_1_, *F*_2_, *F*_3_, *F*_4_, *F*_5_] is the feature vector, *τ* is the elbow torque predicted by the model, *ω_in_* and *b_in_* are the weights and thresholds of the input layer to the hidden layer, and *ω_out_* and *b_out_* are weights and thresholds from the hidden layer to the output layer.

To avoid the neural network falling into local optimums and having a slow convergence, the adding momentum mechanism and adaptive learning algorithm were adopted to improve the accuracy of pattern recognition and convergence speed of the BP neural network. The weight learning formula with additional momentum and the adaptive learning rate formula are as follows:(7)$$\Delta \omega_{\left(k\right)}=\eta_{\left(k\right)}[(1-\mu )g_{\left(k\right)}+\mu g_{\left(k-1\right)}]$$
(8)$$\eta_{\left(k\right)}=2^{\lambda }\eta_{\left(k-1\right)}$$
(9)$$\lambda =sign[g_{\left(k\right)}g_{\left(k-1\right)}]$$
where *k* is the number of training sessions in the network, Δ*ω*_(*k*)_ is the weight increment for the kth training, *g*_(*k*)_ is the gradient calculated for the kth training, *η*_(*k*)_ is the learning rate for the kth training, and *λ* represents the gradient direction.

Particle swarm optimization is used to adjust the initial weight threshold of neural networks in order to further improve the global search capability and increase convergence speed and accuracy. As shown in Table 3, specific parameters were set.

### 2.7. Real-Time Active Torque Control Protocol

Combining the multi-modality information fusion and active torque prediction techniques, we proposed a real-time control technique based on an active torque prediction model, as shown in Figure 8. The initial step was to gather raw EMG signals from the subject’s biceps, triceps, and brachioradialis muscles. By analyzing the amplitude, frequency, and temporal characteristics of EMG signals, it was possible to predict human movement intentions and patterns. And the elbow angle was calculated based on the angles of the upper arm and forearm. The second step involved the data processing and feature extraction of the collected signals to obtain the input features for the active torque prediction model, including elbow angle and angular velocity, and the *RMS* and *iEMG* of the biceps, triceps, and brachioradialis muscles. This eight-dimensional feature vector was converted into a reduced multi-dimensional principal component feature vector via Section 2.4 Multi-Modality Information Fusion in the third step. The fourth step was to predict the active torque by using the active torque-prediction model and calculate the assist torque based on the active torque. The assist ratio, a configurable parameter in this phase, determined the assist torque, which was defined as the active torque multiplied by the assist ratio. The assist torque and assist ratio are directly proportional under the same condition, and the assist ratio value can be adjusted to accommodate the various rehabilitation cycles of stroke patients to provide individualized and precise training for a better rehabilitation effect. The last step is to control the output torque of the electronic commutated (EC) motor in the exoskeleton by converting the auxiliary torque magnitude to the current magnitude. Subjects can move their elbow in the flexion and extension positions with the help of the exoskeleton. The maximum assist torque is predetermined for the user’s safety. The assist torque was adjusted to the maximum value set, while it exceeded the set auxiliary torque range, and constrained the maximum angular velocity of the exoskeleton. Additionally, we adopted the positive and negative angular velocities of the elbow joint to control the direction of the auxiliary torque.

### 2.8. Model Evaluation Index

The root-mean-square error (*RMSE*) and coefficient of determination of the actual and projected values were calculated to assess the accuracy of active joint torque prediction. The closer the *RMSE* is to zero, the lower the error, and it is used to assess the difference between the actual value and the estimated value. The variables that can be described by independent variables through a non-linear regression relationship are known as the coefficient of determination, and the closer it is to 1, the better the fit. The calculation formulas of evaluation indicators are as follows:(10)$$RMSE=\sqrt{\frac{1}{n}\sum_{i=1}^{n}(y_{i}-{\hat{y}}_{i}{)}^{2}}$$
(11)$$R^{2}=1-\frac{\sum_{i=1}^{n}(y_{i}-{\hat{y}}_{i}{)}^{2}}{\sum_{i=1}^{n}(y_{i}-\bar{y}{)}^{2}}$$
where *n* is the sample number of elbow torque, *y_i_* is the actual elbow torque, ${\hat{y}}_{i}$ is the predicted elbow torque, and $\bar{y}$ is the average of the actual elbow torque. In general, models were considered to perform well if the ratio of the *RMSE* to the actual torque mean was less than 15% and the *R*^2^ was greater than 0.9 [36].

## 3. Results

For the purpose of training active torque prediction models, EMG data and joint angle signals of elbow flexion and extension were collected synchronously in this study. Figure 9 and Figure 10 show the EMG signals and angle signals for one cycle. In total, 240 flexion and extension movement data points were collected from each subject for a total 1680 times. To validate the performance of the proposed real-time active torque control model and the consistency of the predicted and actual torque, the experimental data were divided into training, validation, and test groups at a ratio of 6:2:2. The validation set was used to test the overfitting, while the training set was used to train the model for greater accuracy. If the estimation error of the validation set increases as the training accuracy increases, it means that the network is overfitting and the training process needs to be terminated. The test set puts the network’s capacity for generalization to the test when training is finished. Figure 11 shows a comparison plot of the active and actual torque predicted by the trained model. This study uses the second type of Lagrange method to calculate the elbow joint torque of the human upper limb during bending and extension, and uses it as the actual torque. The *RMSE* of the data in Figure 11 was 0.0559 N·m, the actual torque average was 2.3007 N·m, the ratio was 2.43%, and *R*^2^ was 99.67%, demonstrating that the model may be used for online control systems, has strong generalizability, and is not overfitted. The embedded system performed active torque prediction.

The subject’s upper limb’s movement status was continuously changing, and the amount of the auxiliary torque was doing the same. As a result, the entire human–machine system was stable and closed-loop.

We also included the active torque prediction model following training in the real-time control approach in an embedded system. We conducted pertinent experiments to evaluate the active torque prediction model’s performance in embedded systems. In this experiment, the embedded system’s real-time anticipated torque and the parameter values needed to determine the actual torque when the subject performed flexion and extension motions were both collected using a PC. Following analysis, Figure 11 compares the embedded system’s real-time anticipated torque with its actual torque. The *RMSE* of the data in Figure 12 is 0.1956 N·m, the actual average torque is 2.4344 N·m, the ratio is 8.04%, and the *R*^2^ is 94.98%. Even though the data processing techniques used in embedded systems have some limits, the active torque prediction model’s real-time prediction performance was still quite good and perfectly satisfied the requirements for exoskeleton real-time control.

The embedded system’s active torque-prediction model’s real-time performance was further assessed, with the assessment procedure beginning with the initial signal acquisition and ending with the anticipated torque production. The single estimation required roughly 40 ms, of which about 25 ms were spent on data collection, 10 ms on data processing, and the remaining 5 ms on data transmission. Using DMA technology in the master chip, a single chip implemented both data acquisition and data calculation simultaneously in the specified embedded system. The application of direct memory access (DMA) technology in continuous real-time control increased the estimated active torque per second. Owing to the complexity of the actual control program operation, we experimentally evaluated the output frequency of the embedded system’s predicted torque, as depicted in Figure 13. The average output frequency of the torque throughout testing was anticipated to be 31.80 Hz.

## 4. Discussion

We developed an exoskeleton with a single active degree of freedom, only 1.5 kg in weight, that can assist patients with elbow flexion and extension movements, which would be easy enough to use at home, as well as in medical facilities and community centers. Based on the active freedom exoskeleton, a real-time active torque prediction control model was proposed, which aimed to provide auxiliary torque based on the intention of patients and assist patients in completing rehabilitation training so as to improve the treatment effect significantly. By learning and studying user intention-driven rehabilitation robots, we employed the BP neural network with the momentum mechanism and adaptive learning algorithm for training active torque prediction models out of all the available techniques. To increase the model’s robustness, we incorporated multi-source physical and physio-logical signals as model inputs, such as elbow angle signals and EMG signals from several different muscles.

This study proposes a real-time control method for an upper limb exoskeleton based on an active torque-prediction model. The model not only worked well with offline data on a PC, but it also worked well with embedded systems’ real-time torque prediction. According to the thorough investigation, the embedded system’s real-time prediction torque root-mean-square error was 0.1956 N·m. The upper limb torque was also researched by Diana R. Bueno et al. They used the Hill model to predict the upper limb active torque and the data analysis revealed that the elbow active torque prediction’s *RMSE* was approximately 1.4 N·m [37]. It is evident that the model put forward in this research had a low *RMSE*.

Time delay is one of the crucial factors that cannot be overlooked in an exoskeleton real-time control system. In this study, the embedded system’s primary control chip was the STM32F407ZGT6, and its primary control frequency was 168 MHz. The embedded system only needed 40 ms to evaluate a single active torque, including 10 ms for data processing and 25 ms for data collection. In the same type of study, the real-time training system for upper limb exoskeleton robots based on EMG signals developed by Gao et al. had an average delay time of 300 ms, while the control approach suggested by Ye et al. had a delay time that was almost as long [2,38]. The model proposed in this study clearly had minimal delay.

Delays shorter than 300 ms are typically acceptable in EMG control because the human body cannot detect them [38]. However, the flexibility of human–computer interactions improves and patient experience improves with decreasing time delays. When calculating eigenvalues, this study adopted the form of non-overlapping time windows, which meant that there was no overlapping data between adjacent time windows, and each time window’s data could calculate an eigenvalue. The sampling frequency of the surface electromyography signal in this article was set to 1200 Hz, which meant that 1200 data points were collected per second. A time division window of 25 ms was selected, and the moving step was also 25 ms, which meant that a root-mean-square value will be calculated for every 30 surface electromyography signal values. The main factor contributing to the proposed model’s minimal latency was the ability to predict active torque with less data. The active torque prediction results were still good, despite the sparse data, which further demonstrated how well this model performed.

Depending on how well a patient has recovered from their stroke, the course of treatment is divided into various phases. When the patient’s upper limb recovers steady EMG signals, the exoskeleton control approach described in this work can be initiated [20]. The exoskeleton real-time control approach is intended to let individuals with varied degrees of impairment move their elbows. The motor torque and the active torque produced by the weaker muscles together make up the actual torque required to rotate the forearm. The degree of limb dysfunction determines how much auxiliary torque the motor produces. In fact, less active torque and lower EMG signals were generated by weaker patients. As a result, they need motors to produce greater auxiliary torques, which increases the amount of amplification needed for their anticipated active torques. This can be managed by modifying the variable auxiliary ratio, which, under similar circumstances, is proportionate to the auxiliary torque and auxiliary ratio. Additionally, until patients are able to move their forearms independently, this will enable them to use the same system [16].

This study may have a flaw in that surface electrode recordings of EMG signals may be affected by minute relative movements between the skin and muscle, which are especially noticeable at the biceps brachii muscle. In the future, we will further explore alternative electrode types and configurations that can provide better stability and reduce the impact of relative motion, as well as other measures to stabilize electrode placement. Furthermore, as the trial was conducted on healthy volunteers, more research is required to determine how the results would apply to patients.

## 5. Conclusions

In this study, we developed an exoskeleton real-time control approach based on a prediction model of active elbow joint torque using the BP neural network. In order to forecast the elbow joint active torque, we first acquired the EMG signal and elbow joint angle of the relevant upper limb muscles, processed the data to extract the necessary features, and then used the relevant features as input parameters. Additionally, we determined the auxiliary torque based on the active torque projected in real-time and delivered the associated auxiliary torque by regulating the servo motor’s current magnitude. EMG signals added to the source of the control signal enhanced not only the patient’s motivation for rehabilitation, but also the ability of human–computer interactions. Additionally, utilizing offline data from a PC and real-time data from embedded systems, we examined the performance of the active torque prediction model. According to the findings, the model’s accuracy was quite high and its real-time testing latency was extremely low, fully satisfying the demands of practical application.

## Figures and Tables

**Figure 1 bioengineering-10-01441-f001:**
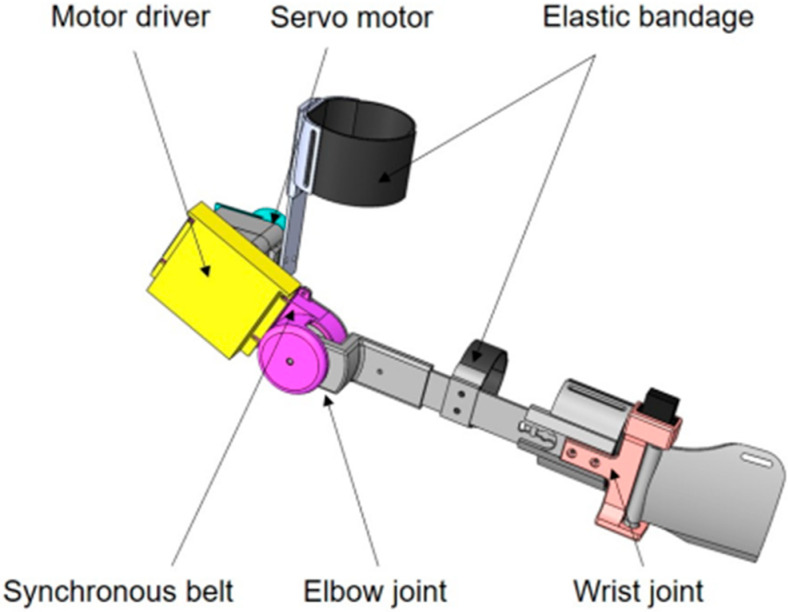
The three-dimensional model of the exoskeleton.

**Figure 2 bioengineering-10-01441-f002:**
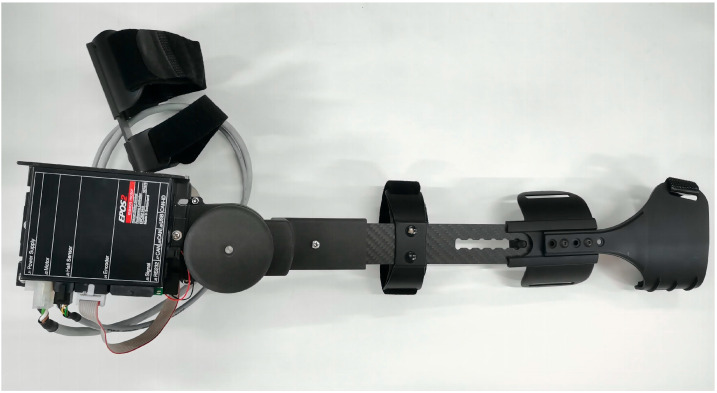
Exoskeleton prototype.

**Figure 3 bioengineering-10-01441-f003:**
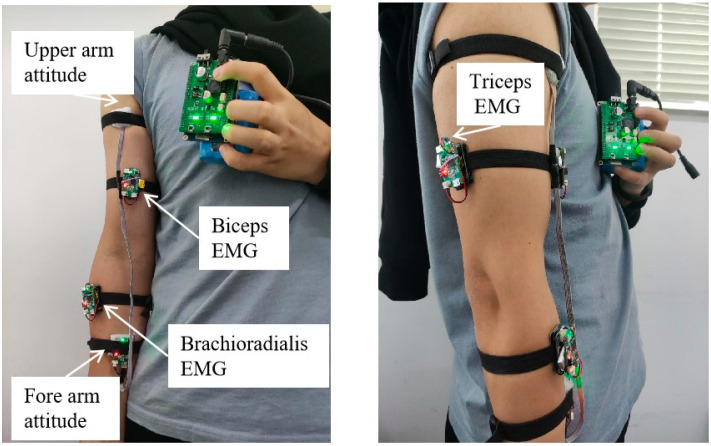
The positions of the sensors.

**Figure 4 bioengineering-10-01441-f004:**
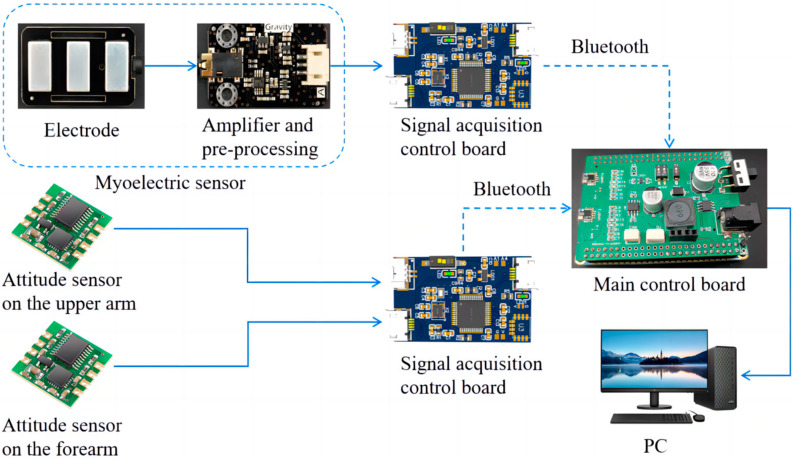
The data acquisition system.

**Figure 5 bioengineering-10-01441-f005:**
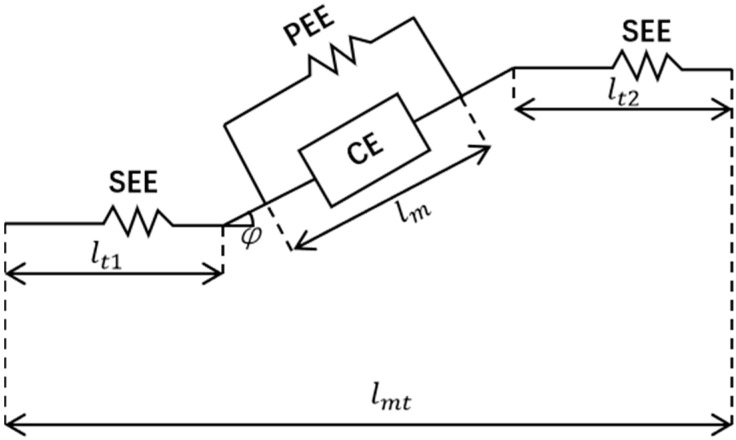
Hill muscle model.

**Figure 6 bioengineering-10-01441-f006:**
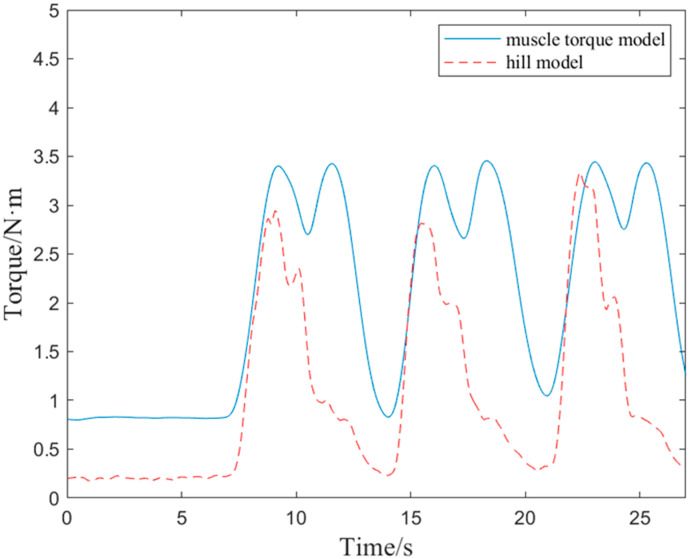
Comparison of results between muscle torque model and hill model.

**Figure 7 bioengineering-10-01441-f007:**
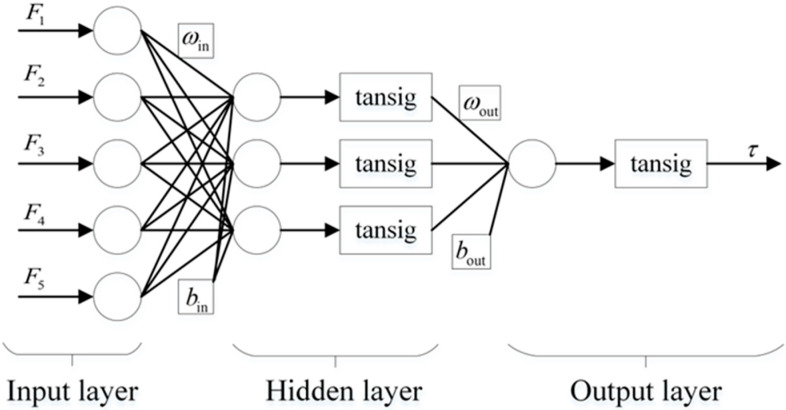
The structure of the BP neural network.

**Figure 8 bioengineering-10-01441-f008:**
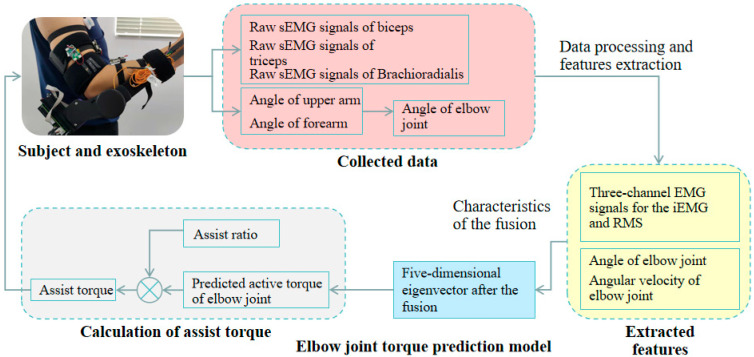
Real-time control method flow diagram.

**Figure 9 bioengineering-10-01441-f009:**
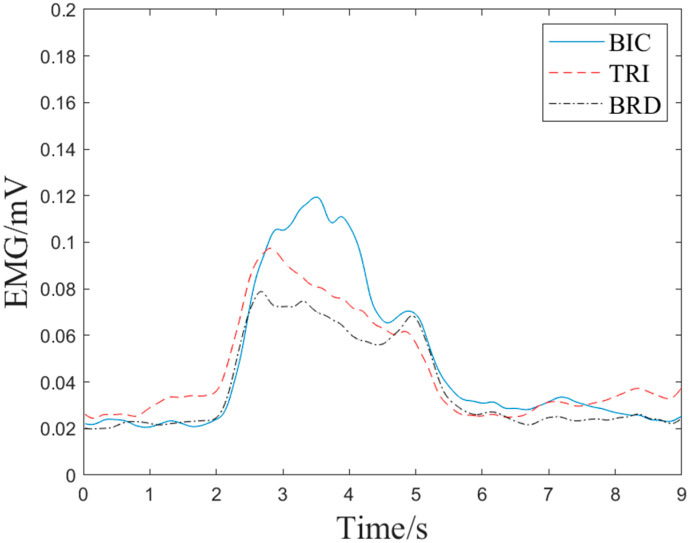
EMG data of elbow flexion and extension.

**Figure 10 bioengineering-10-01441-f010:**
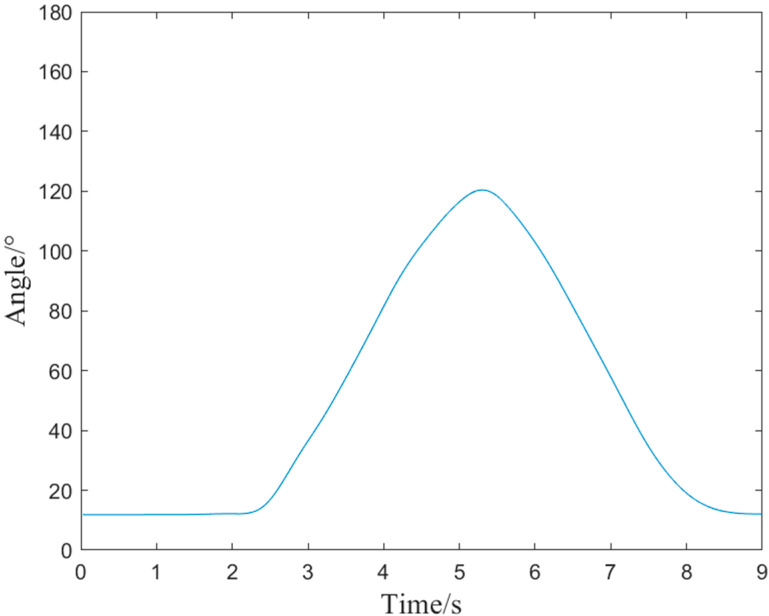
Joint angle signals of elbow flexion and extension.

**Figure 11 bioengineering-10-01441-f011:**
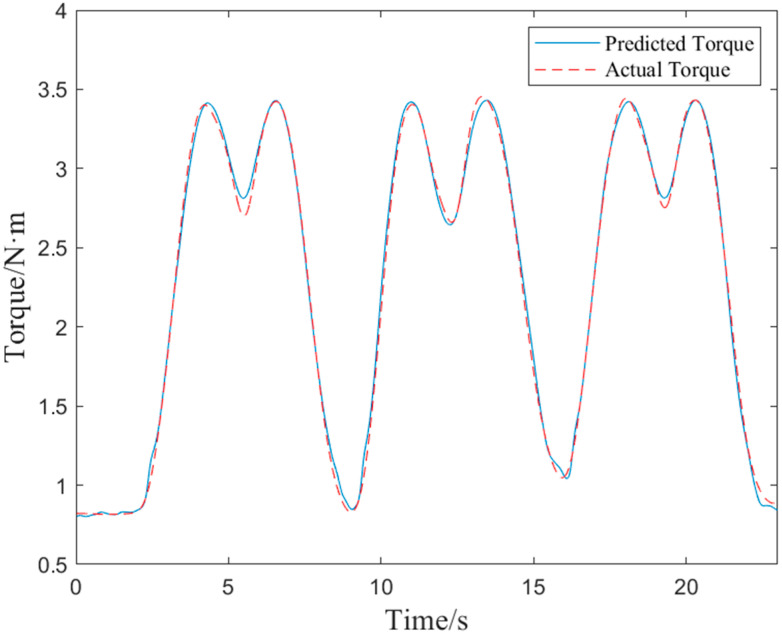
Test data-predicted torque vs. actual torque.

**Figure 12 bioengineering-10-01441-f012:**
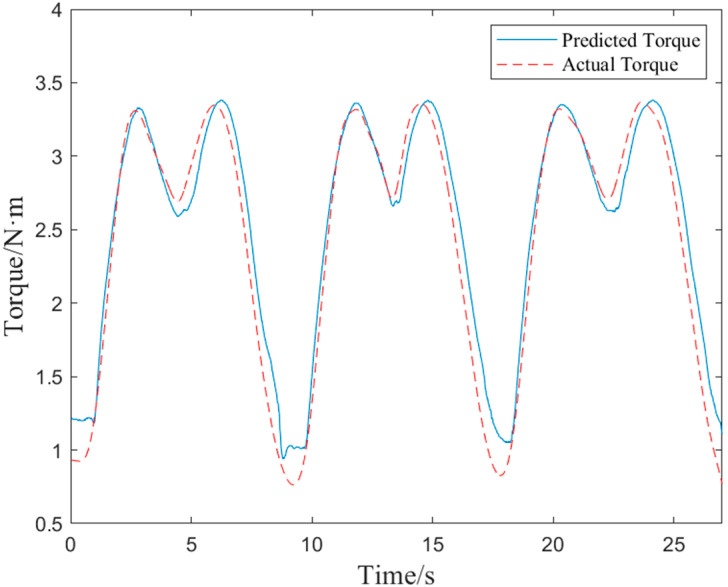
Predicted versus actual torque for embedded systems.

**Figure 13 bioengineering-10-01441-f013:**
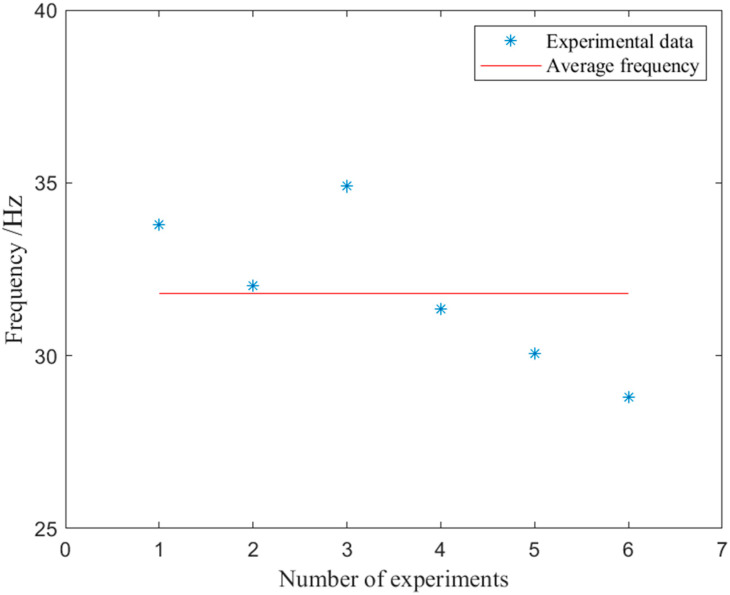
Output frequency of predicted torque for embedded systems.

**Table 1 bioengineering-10-01441-t001:** Subject anthropometric information log.

Subject	Age	Height (cm)	Weight (kg)	Upper Arm Length (cm)	Forearm Length (cm)	Hand Length (cm)
1	27	165	70.2	26.0	25.0	18.0
2	26	175	60.5	35.8	26.2	19.1
3	26	180	79.5	33.4	27.3	19.0
4	28	178	53.5	30.6	27.9	19.4
5	24	166	64.4	28.6	26.0	18.1
6	25	172	70.0	29.6	27.0	18.7
7	24	167	65.3	28.7	26.2	18.2
Mean ± SD	26 ± 2	173 ± 7	66.5 ± 13	30.9 ± 4.9	26.45 ± 1.45	18.7 ± 0.7

SD means standard deviation.

**Table 2 bioengineering-10-01441-t002:** Each feature vector’s contribution rate and cumulative contribution rate.

Principal Component	Feature	Contribution Rate/%	Cumulative Contribution Rate/%
F1	6.4999	81.25	81.25
F2	1.0455	13.07	94.32
F3	0.2404	3.01	97.32
F4	0.1251	1.56	98.89
F5	0.0878	1.10	99.98
F6	0.0006	0.01	99.99
F7	0.0003	0.00	100.00
F8	0.0003	0.00	100.00

**Table 3 bioengineering-10-01441-t003:** Specific parameters.

Population Size *N*	Learning Factor *c*_1_	Maximum Iterations *t_max_*	Linear Decrement in Inertial Weight	
			Initial Value *ω_max_*	Final Value *ω_max_*
100	2	2000	0.9	0.4

## Data Availability

Data are contained within the article.

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
