# Peer review of "A Real-Time Control Method for Upper Limb Exoskeleton Based on Active Torque Prediction Model"

_bioengineering, 2023, doi:10.3390/bioengineering10121441_

Round 1

Reviewer 1 Report

Comments and Suggestions for Authors

Dear Authors,

I suggest you to consider the following comments / suggestions for improving your manuscript:

In my opinion, this paper must be the subject of extensive reworked since it is not very clear and it is very difficult to read, follow and understand.

One of the main problems, it that even the subject of the paper is not very clear. For instance, on the abstract it is stated: " This research proposes a real-time active torque control system for upper limb exoskeletons based on muscle force magnitudes."

However, on the Discussion section, it is stated: "This study proposes and evaluates a prediction model for the elbow joint's active torque. The model not only works well with offline data on a PC, but it also works well with embedded systems' real-time torque prediction".

Additionally, the authors mentions along the text several different torques and each one's meaning should be clearly presented upfront.

Furthermore, the paper should be subjected to an extensive language review since some sentences are not very clear and make it difficult to clearly understand what the authors intend to express.

Was there any informed consent on the part of the seven subjects used for the tests? This is a requirement of tests performed with live human subjects. Were the experiments subjected to, and approved by, a ethics committee?

Some more detailed comments follow:

- Does the sentence "Some essential drive control models used in the context of team members’ exoskeleton mechanism are briefly reviewed and discussed", on page 2, mean that are only reviewed previous works from the same teams as the authors? 

-Page 3: check the sentence: "The range of flexion ... displays in Figure 1"; this sentence is very long and difficult to understand. Avoid using such long sentences.

- Page 4: I understand, from the sentence "The self-developed EMG ... upper limb exoskeleton." that the EMG and attitude sensors used were developed by the authors. If this is the case, briefly describe these sensors or include a bibliographic reference to a document describing its characteristics. If commercial sensors are used, then present their brand and model and include a link to the manufacturer.

- Page 4: what are the technical details of the "...signal collecting control panel"? See previous comment.

- Page 4: the sentence "The main control board ... active elbow torque " is not clear to the reviewer. Consider rewriting it so that its meaning is comprehensible to readers.

- Figure 3: what is the brand / model of the used boards?

- Page 5: It is not clear why the number of segments is N=30 - please explain.

- Page 5: where it is mentioned "Thus, this work extracted angular velocity ... robustness of the control system", it is not clear how this information is extracted. Is it through the attitude sensors? Make a clear explanation of this.

- Page 7: consider including a bibliographic reference to the "second type of Lagrangian method" or further describe what is meant with this statement - it is not very clear.

- Page 7: the meaning of sentence "The overall results ... relatively large" is not clear to the reader; consider rewriting it.

- Page 9: sentence "...the amplitudes extracted from which determined the subject's intention." is not clear to the reader; consider rewriting it.

- Page 9: improve the resolution of Figure 7.

- Page 9: regarding Figure 9, how are calculated the angles of the upper arm and of the forearm? It is is with the attitude sensors, I believe it is needed a calibration procedure, and this is not presented in the paper. Please make this clear.

- Page 11 and Figure 10: it is not totally clear how the actual torque was determined / computed.

- Page 14: It is not totally clear how the embedded system performs the active torque prediction and which model is used for this purpose.

- Page 15: the information "The window length is 25 ms, and the moving step length is 25 ms. Each window has 30 data points." should had been presented before, to fully describe the methods used.

- The bibliographic references used by the authors are not the most adequate one. Several references are not made to the source works but to works that cite the source works. As an example, on page 5, the sentence "The sampling theorem states that ...as their effective frequency [25]." makes a reference to [25] but it should make a reference to a document, for instance, in signal processing, describing Nyquist Theorem.

- According to the mathematical notation, all variables should be typed in italics font. Check and correct along the entire manuscript.

- According to the International System of Units, there should be a space between the value an the unit. For example, on page 5, where it is "500Hz", it should be "500 Hz". Correct this problem along the entire manuscript.

- All acronyms presented on the document should be defined:

Page 2: Define acronym BP

Page 10: Define EC acronym

Page 14: define DMA acronym

Finally, check and correct the following typos:

Page 2: "signals[4]" -> "signals [4]"

Page 4: "...respon-sible or controlling ..." -> "...respon-sible for controlling..."?

Page 5: "[0xb0]" ?

Page 8: "...adding the number of hidden layers..." -> "...increasing the number of hidden layers..."?

Comments on the Quality of English Language

Dear Authors,

The paper should be subjected to an extensive language review since some sentences are not very clear and make it difficult to clearly understand what the authors intend to express. Please see my detailed comments above.

Reviewer 2 Report

Comments and Suggestions for Authors

This paper developed an upper limb exoskeleton as the rehabilitation robots. A real-time active torque control system was adopted for this upper limb exoskeletons. This work is quite meaningful in the biomedical engineering. This paper is well-organized and well-written. I recommend that it can be accepted after minor revision.

1. The author claimed that the delay of this system is extremely low with value of only 40 ms. The author should try to summarize the delay from previous studies to compare the delay.

2. In figure 1, a 3D model of the exoskeleton was demonstrated. Actually, a picture of the manufactured exoskeleton also needs to be shown in this paper.

3. To demonstrate the practicality of this exoskeleton, a movie is needed during the actuation process of the active torque control system.

Reviewer 3 Report

Comments and Suggestions for Authors

I recommend that the authors try to revise the structure of the article to improve readability.

Much of the introduction is devoted to previous studies on user intention-driven rehabilitation robots, which I would have expected to be recalled in the discussion section. Try to emphasize more on the originality of the work, what features were improved in this work compared to the literature described in the introduction.

The section Materials and Methods is very long also because it includes digressions (see for example the part on Hill muscle model or that on the artificial neural network), which make it difficult to understand what parts are literature and what was used or developed in the paper instead. For instance, it is not clear to me what Figure 5 should represent. Is it a comparison between the literature Hill muscle model and the muscle torque model used by the authors? To be honest, I do not agree that the two models provide "basically the same result" (line 232)

I find paragraph 2.4 on Multi modality information fusion very difficult to understand, especially regarding the PCA analysis and Table 1. What information in this section is relevant to readers in order to better understand the “real time control method” developed in the paper? If relevant I suggest Authors provide more detail on the principal components (even just F1 and F2 which alone account for 94.32%). Principal components are linear combinations of parameters, or "summary indices," as they are sometimes called. What combination of the parameters (elbow angle, angular velocity, iEMG and RMS of the three-channel EMG signals) are F1 and F2?

Details on the subject information (3.1) should be moved to the Materials and Methods Section (as they are not results), together with any information concerning the experiments.

In contrast to Materials and Methods, the Results and Discussion Sections are very limited. I recommend that the Authors provide more comments to help readers understand the different figures. It is not straightforward to understand the data in Figures 8 and 9; it probably would have been clearer to plot the EMG signals with reference to joint angle.  It is also difficult for me to judge whether the scatter of the frequency data shown in Figure 12 (which does not seem small) might be acceptable and what the implication of "calculating an average frequency" is.

In general, I recommend that the authors expand lines 432-335 and discuss the limitations of their work, especially with respect to how experimental data on healthy subjects might provide useful data with respect to a use of exoskeleton and control logic on post-stroke subjects.

Comments on the Quality of English Language

I recommend that the authors have the article read by someone who knows English well. In several places in the text, the verb form and  prepositions used do not seem correct. This makes it very complex to understand what the authors mean. 

Online translators can help but sometimes , especially in scientific contexts, fail to translate the technical concept correctly. 

Reviewer 4 Report

Comments and Suggestions for Authors

This manuscript discussed a real time control method for upper limb exoskeleton, centered on an active torque prediction model, with a low delay of 40 ms obtained. The topic is of interest for this journal, it is the opinion of this reviewer that the manuscript can be considered acceptable for publication after addressing the comment below.

1.      In Figure 12, it is unclear how to determine the average output frequency of the torque. There is a significant discrepancy between experimental testing results and predicted data. Please clarify.

2.      Could the potential impact of minute relative movements between the skin and the biceps brachii muscle on surface electrode recordings of EMG signals be further addressed or mitigated in the study's methodology or discussion? Please clarify.

Comments on the Quality of English Language

Minor

Round 2

Reviewer 1 Report

Comments and Suggestions for Authors

Dear Authors,

Congratulations for your work.

Warm regards,

Manuel